# Functionalization and higher-order organization of liposomes with DNA nanostructures

Zhao Zhang [1] ✉, Zhaomeng Feng [1], Xiaowei Zhao[2], Dominique Jean[2], Zhiheng Yu [2] & Edwin R. Chapman [1] ✉

Small unilamellar vesicles (SUVs) are indispensable model membranes, organelle mimics, and drug and vaccine carriers. However, the lack of robust techniques to functionalize or organize preformed SUVs limits their applications. Here we use DNA nanostructures to coat, cluster, and pattern sub-100-nm liposomes, generating distance-controlled vesicle networks, strings and dimers, among other configurations. The DNA coating also enables attachment of proteins to liposomes, and temporal control of membrane fusion driven by SNARE protein complexes. Such a convenient and versatile method of engineering premade vesicles both structurally and functionally is highly relevant to bottom-up biology and targeted delivery.

Liposomes are biomimetic analogs of natural membranes, typically consisting of phospholipid bilayers enclosing aqueous media. Their excellent versatility and biocompatibility have led to applications in analytical science[1], drug and vaccine delivery[2], synthetic biology[3], etc. Unleashing the full potential of liposome-based products relies on efficient surface modification and precise interaction control[4,5]. Although existing techniques are capable of coating and associating giant unilamellar vesicles (GUVs)[6], generic tools for functionalizing and patterning liposomes with diameters smaller than 100 nm have not been established. These small unilamellar vesicles (SUVs) not only serve as commercial drug carriers[7], but also simulate important organelles such as synaptic vesicles (SVs) in size and composition[8], thus being essential for both in vivo and in vitro studies.

DNA nanotechnology is a powerful method for membrane engineering[9,10]. Previously, simple DNA constructs, like single-stranded (ss) or double-stranded (ds) DNA, were interfaced with membranes to direct the assembly/disassembly of liposome aggregates, and to drive vesicle fusion, in a programmable manner[11,12].

Here we aim to expand this toolbox by introducing more sophisticated DNA architectures[13,14], to achieve rigorous and efficient manipulation of both membrane structure and function[15-17]. A variety of self-assembled DNA nanostructures were first anchored on membrane surfaces, then used to dictate protein display and higher-order organization of liposomes. Liposome-nanodisc complexes, distance-controlled liposome networks and duos, and one-dimensional (1D) liposome arrays were created among other conformations. DNA-mediated coating and clustering was also employed for temporal control of membrane fusion driven by soluble N-ethylmaleimide-sensitive factor attachment protein receptor (SNARE) complexes[18]. These examples open the door for new and more advanced liposome applications in the future.

## Results

### Protein display on liposomes

Liposomes used in this study were made of 65% 1,2-dioleoyl-sn-glycero-3-phosphocholine (DOPC), 15% 1,2-dioleoyl-sn-glycero-3-phosphoethanolamine (DOPE), and 20% 1,2-dioleoyl-sn-glycero-3-phospho-L-serine (DOPS), unless otherwise specified. Liposomes were prepared by dialysis-based detergent removal followed by buoyant density centrifugation (See Supplementary Materials and Methods). Preformed liposomes were mixed with 5′-modified cholesterol-ssDNA (DNA-chol) and incubated at 30 °C for 30 min, to allow hydrophobicity-driven membrane anchoring by DNA[19]. The resultant ssDNA-tethered liposomes would readily enable the attachment of DNA nanostructures bearing complementary ssDNA extensions (handles), in this case a six-helix-bundle (6HB) DNA stick (tile) with three handles at one end (see Supplementary Fig. 1 for design blueprint, Supplementary Fig. 2 for gel validation). Products were characterized

[1]Howard Hughes Medical Institute, Department of Neuroscience, University of Wisconsin–Madison, Madison WI 53705, USA. [2]Howard Hughes Medical Institute, CryoEM Shared Resource, Janelia Research Campus, 19700 Helix Drive, Ashburn VA 20147, USA. ✉e-mail: zhao.zhang@wisc.edu; chapman@wisc.edu

by negative-stain transmission electron microscopy (TEM) as shown in Fig. 1. The number of handles on each tile was critical to the yield and stability of tile binding, where at least two handles were required for efficient coating (Supplementary Fig. 3)[20]. No attachment was observed for tiles without handles or with three non-complementary handles, ruling out the presence of nonspecific binding due to electrostatic interactions in the current system (Supplementary Fig. 3)[21]. A 100:1 molar ratio of lipids to DNA-chol was appropriate for tile-liposome association (Supplementary Fig. 4), and thus was used in this study.

Coating liposomes with DNA tiles paved the way for follow-up functionalization with various molecules of interest (MOI). To achieve this, a handle with a different DNA sequence was included at the other end of the 6HB tile. This handle can hybridize with its complementary oligonucleotide pre-conjugated to the MOI. In Fig. 1 we showcased two such proofs of concept, both displaying proteins on liposomes. The first example used streptavidin (SA) bound to biotinylated ssDNA (DNA-biotin) on the tiles, as a representative of soluble proteins. Negative-stain TEM images showed particles with a liposome core studded with tile-protein complexes, agreeing with the design (Fig. 1 right top and Supplementary Fig. 5). The second example demonstrated functionalization with membrane proteins: synaptobrevin-2 (v-SNARE) was first reconstituted in a DNA-tethered nanodisc (DNDv, see Supplementary Fig. 6 for gel and TEM characterization)[22], then bound to the periphery of the liposomes. In negative-stain TEM images, liposomes surrounded by a crown of nanodiscs (NDs) were observed (Fig. 1 right bottom and Supplementary Fig. 7), presenting a new type of hybrid model membrane system. The role of DNA nanostructures here was to provide handles and spacing for liposome-MOI association, with the potential of enabling precise control over distance[23], stoichiometry[24], and dynamics[25], all of which are highly important for biomimicry and for the design of regent delivery vehicles.

## Distance-defined liposome clusters

If a 6HB tile with handles at one end could anchor to a ssDNA-tethered liposome as shown in Fig. 1, we predict that the same tile, with handles at both ends, should be able to unite two such liposomes together.

This hypothesis was tested and verified by negative-stain TEM studies, where liposome clusters spanning up to several micrometers were found after incubation of tiles with liposomes at 30 °C for 1 h (Fig. 2a and Supplementary Fig. 8). Stick-like tile structures ~30 nm in length could be discerned between neighboring liposomes in close-up views, further supporting the scheme. Cryogenic electron tomography (cryo-ET) was also used to characterize tile-mediated liposome clusters (Fig. 2b and Supplementary Fig. 9). Compared to potential sample deformation during negative staining, samples were rapidly frozen in vitreous ice for cryo-ET without dehydration or staining, which preserved the 'native' shape and configurations. As expected, intact and spherical liposomes interconnected by straight tiles were observed, confirming that the attaching and linking by DNA nanostructures didn't impair membrane morphology. The measured membrane-to-membrane distance (29 ± 2 nm) matches the length of the linker tile (30 nm) well. 3D modeling was performed to highlight the components (liposomes and tiles) and topology of the cluster (Fig. 2b).

The size of liposome clusters can be regulated by tuning the ratio of lipids to DNA-chol, as well as by varying the concentration of tiles. In general, more DNA-chol and/or tiles led to bigger agglomerations (Supplementary Fig. 10). Furthermore, DNA-linked liposomes could be separated in response to specific nucleic acids, driven by toehold-mediated strand displacement (TMSD)[26]. As demonstrated in Fig. 2c, monodispersed liposomes reappeared after the corresponding displacing strand or DNase I was added, and detached DNA tiles were visible in the background in the former case. It is worth noting that the size distribution of individual liposomes before and after disassembly remained unchanged, indicating that minimal fusion occurred during clustering (Supplementary Fig. 10). Previously, reversible vesicle aggregations have been achieved on the basis of biotin-SA binding[27] and copper(iminodiacetate)-histidine interaction[28], among others. But the selective DNA base-pairing provided the ultimate solution for programmable dynamic control[29,30]. Our method could also serve as a model system to study the clustering and release of SVs in presynaptic terminals, which play critical roles in regulating neurotransmission and synaptic plasticity[31].

Another significant advantage of using DNA nanostructures to associate liposomes is their unprecedented designability. While simple

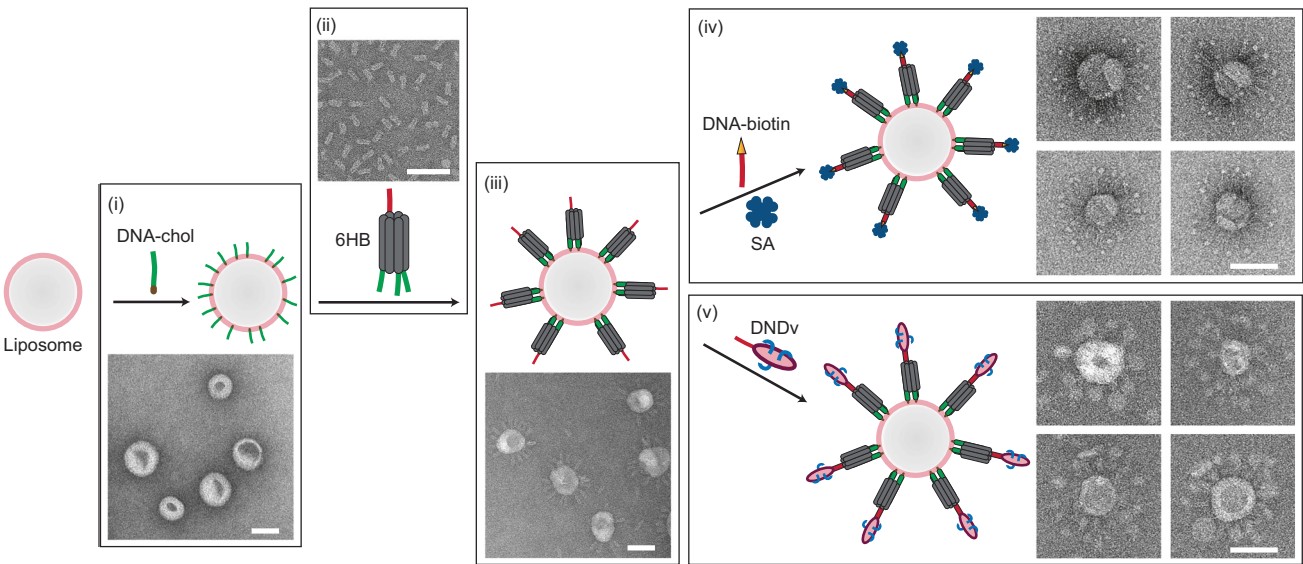

**Fig. 1 | Coating liposome with DNA tiles for protein display.** Preformed liposomes are first modified with ssDNA (i, green line) by cholesterol-membrane association, then coated with 6HB DNA tiles (ii) by DNA hybridization (iii). An additional ssDNA handle on each tile (ii, red line) is reserved for further functionalization with proteins, such as streptavidin (SA) by biotin-SA interaction (iv), or v-SNARE by DNA-tethered nanodisc (DNDv) reconstitution (v). Both liposomes and NDs contain 65% DOPC, 15% DOPE, and 20% DOPS. Representative negative-stain TEM images are shown beside the cartoon model of each construct. Scale bars: 50 nm.

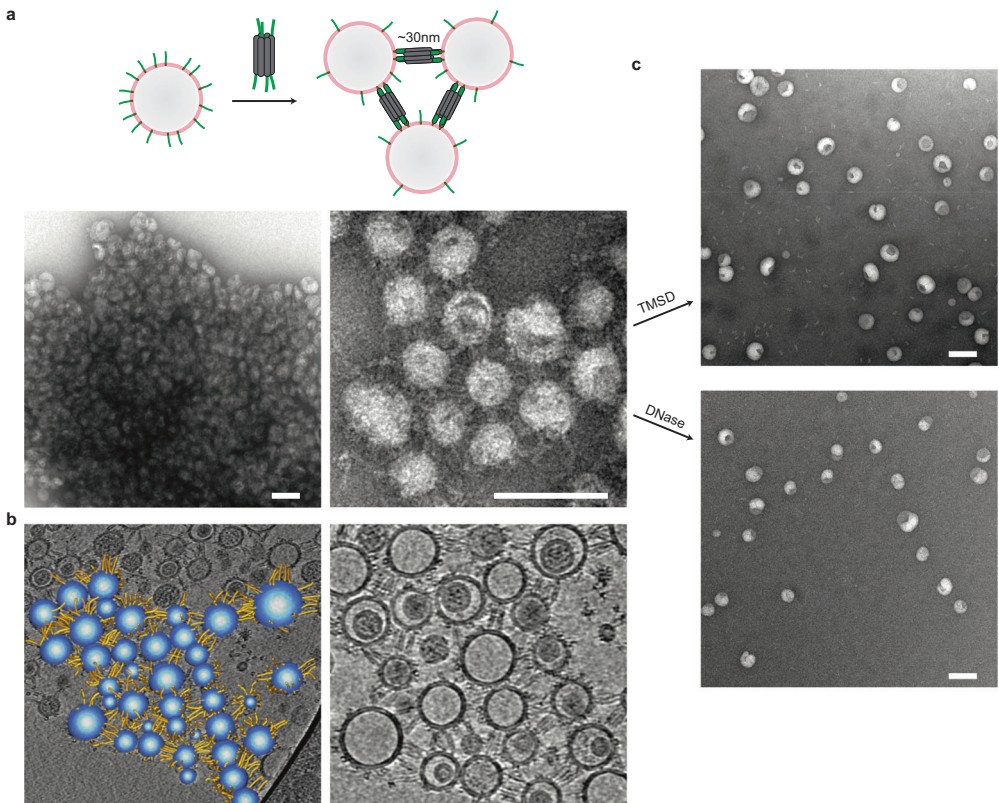

**Fig. 2 | Reversible clustering of liposomes with 6HB DNA tiles. a** Cartoon model and negative-stain TEM images of ssDNA-tethered liposomes clustered by 6HB DNA tiles with handles on both ends. **b** Cryo-ET modeling of tile-coated liposomes. **c** Negative-stain TEM images of disassembled liposome clusters, driven by either toehold-mediated strand displacement (TMSD) or DNase degradation. Scale bars: 100 nm.

ssDNA and dsDNA were also capable of producing liposome aggregates (Supplementary Fig. 11)[32,33], sophisticated DNA constructs like tiles and origami offered more robust and versatile structural features. In this study, we focused on controlling the inter-liposome distance. The standard 6HB tile with handles on both ends had a total length of ~30 nm (Fig. 2a). By switching the handle positions from the ends to the sides of the tile, liposome clusters with smaller intervals (~20 nm) were formed (Fig. 3a). To further shorten the intervals, we coated the liposomes with 3'-modified DNA-chol, so that the tiles could be brought more proximal to the liposome surface with a 'zipper' conformation, theoretically resulting in <10 nm gaps (Supplementary Fig. 12). Negative-stain TEM images revealed tightly associated liposomes, in accordance with our design.

In pursuit of longer inter-liposome distances, we designed and assembled two DNA origami structures with three handles on each of the two opposite sides (see Supplementary Fig. 1 for design blueprint and Supplementary Fig. 13 for gel and TEM images)[34], expecting them to connect and hold vesicles apart by about 60 and 180 nm respectively. Indeed, TEM studies characterized liposome networks up to several micrometers long, within which linker DNA nanostructures lay between liposomes (Fig. 3b, c). Lower concentrations of origami resulted in smaller clusters, where triangular liposome patterns were observed and enriched (Supplementary Fig. 13). Small clusters also allow us to better distinguish individual liposomes and linkers, such that inter-membrane distance within a cluster can be measured in negative-stain TEM images. The results confirmed the trend of longer DNA nanostructures corresponding to larger inter-membrane distances, but in all cases the average distance was 12–15 nm shorter than design (Fig. 3d). We attributed this discrepancy partly to vesicle flattening (Supplementary Fig. 14). A hexagon-shaped DNA origami was also assembled and used for liposome clustering, but the inter-

membrane distance was not very well maintained probably due to its lower rigidity (Supplementary Fig. 14). Taken together, we assembled liposomes into clusters with well-defined interspacing ranging from <10 to ~180 nm. These unique products and methods should appeal to bottom-up biology and provide new tools for cell adhesion[35,36].

## Higher-order organization of liposomes

Cellular functions rely on coordinated organization and communication of membrane-bound organelles. To further recapitulate such complexity, after the success of creating 3D liposome clusters, we took the challenge of breaking the symmetry of spontaneous assembly, to order SUVs in predefined patterns. We planned and executed two strategies, both using hierarchical DNA architectures as templates for vesicle placement[13].

The first strategy is based on tile-assembled lattices. Here, we took a minimalist approach, by using just one oligonucleotide (called C-tile) which was previously shown to polymerize into tubes via thermal annealing[37]. By adjusting the annealing protocol to accommodate liposomes, micrometers-long, sub-100-nm-wide DNA ribbons were generated, without interaction with liposomes (Supplementary Fig. 15). Then we attached a cholesterol to the 3'-end of the C-tiles (C-chol) and mixed them with SUVs. Interestingly, liposomes were arranged into 1D arrays after an overnight annealing (Fig. 4a). Although the underlying DNA structures were not discernible in negative-stain TEM images, the linear array of tightly packed liposomes demonstrated the template effect of tile assembly. Further evidence came from the products when modified and unmodified C-tiles were mixed in a 1:6 molar ratio, where DNA ribbons under discontinuous liposome strings were exposed (Supplementary Fig. 15).

Our second strategy capitalized on higher-order assembly between DNA tiles and origami[13]. Liposomes were first coated with 6HB

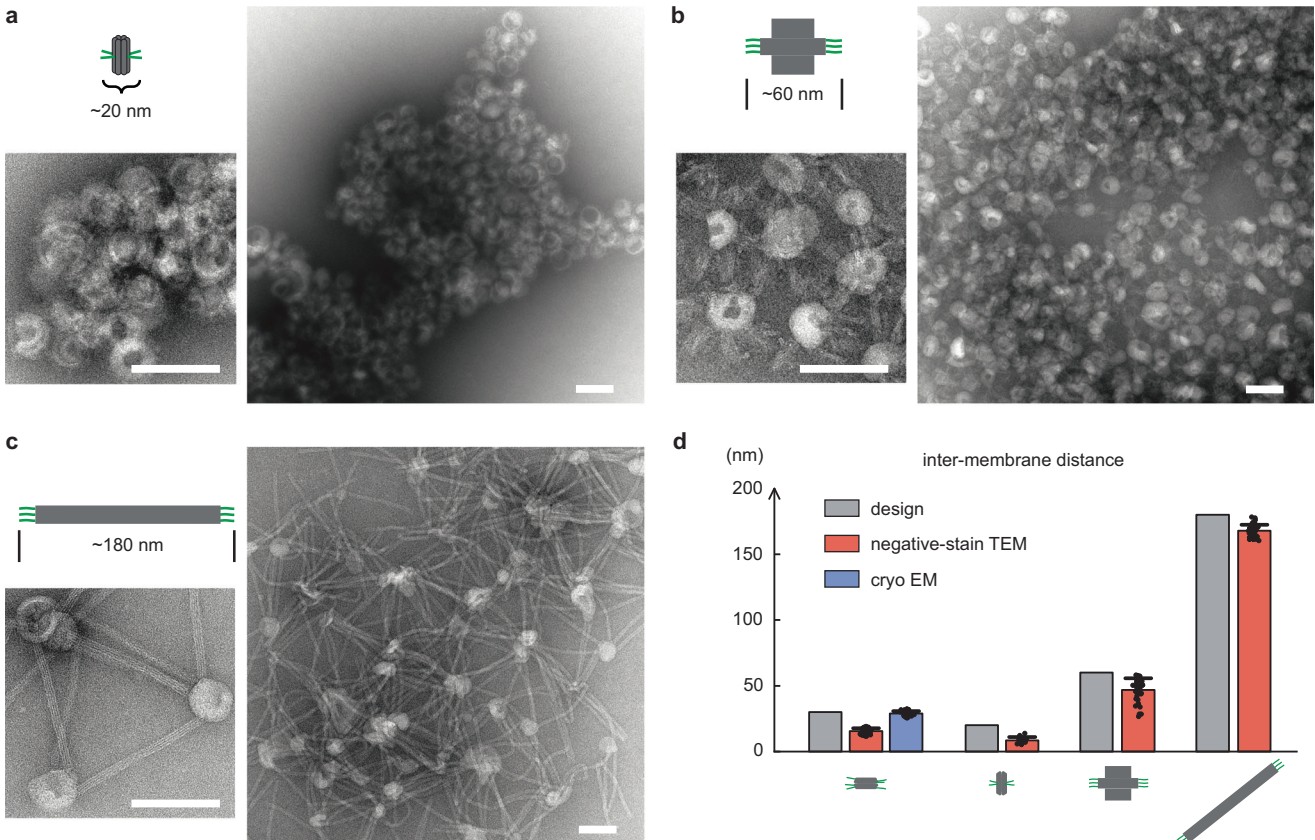

**Fig. 3 | Distance-controlled liposome clusters mediated by customized DNA nanostructures. a–c** Representative negative-stain TEM images of liposomes interconnected by 6HB tiles with side handles (**a**), 60-nm origami brick (**b**), and 180-nm origami rod (**c**). Cartoon model and theoretical full length of each DNA structure is shown on the left of each panel. Scale bars: 100 nm. **d** Bar graph of inter-membrane distances measured in negative-stain (orange) or cryo (blue) EM images.

$N = 47, 16, 34, 40$ for negative-stain EM in each case from left to right. $N = 29$ for cryo EM. Source data are provided as a Source Data file. Individual data points are overlaid with corresponding bar chart. Center and top of error bar represent mean and standard deviation. The cryo measurement matches the design (gray) well, while the negative-stain measurement is shorter than the design partly due to vesicle flattening on substrate.

tiles containing handles pointing outwards, then allowed to hybridize with complementary handles on an origami rod (Fig. 4b). Unlike the 1D liposome arrays which were still a form of periodic patterning, an origami template with prescribed handle positions enabled fully addressable, programmable, and finite assembly. In Fig. 4b, we presented three constructions of liposome dimers with signature distances and configurations. Uniform two-liposomes-one-rod complexes were observed in each case, with tiles visible at the interface, and the measured inter-membrane distances were consistent with the designs (Supplementary Fig. 16). It should be stressed that our method is applicable for pre-existing liposomes, while generation of ordered liposome strings or dimers in previous studies required vesicles to be formed within DNA frames in situ[38,39]. These two strategies could be viewed as complementary techniques for organizing vesicles at different stages of formation.

**Temporal control of SNARE-mediated liposome fusion**
After having demonstrated the power of DNA for coating, clustering, and patterning liposomes, we next harnessed it to apply to membrane dynamics. Although DNA hybridization itself has been exploited to drive fusion[40], in this study we focused on SNARE-driven vesicle fusion, which is an universal and essential biological process in various organisms[41]. A lipid mixing assay was used to characterize fusion kinetics[42]. In this assay, vesicular proteins (i.e. synaptobrevin-2) were reconstituted into donor liposomes (v-liposome or v) containing 1.5% 1,2-dioleoyl-sn-glycero-3-

phosphoethanolamine-N-(lissamine rhodamine B sulfonyl) (Rhod-PE) and 1.5% 1,2-dioleoyl-sn-glycero-3-phosphoethanolamine-N-(7-nitro-2-1,3-benzoxadiazol-4-yl) (NBD-PE), while target-membrane proteins (i.e., syntaxin-1A and SNAP-25B) were reconstituted into acceptor liposomes (t-liposome or t). Fusion between v-liposomes and t-liposomes should lead to the rise of NBD emission, due to decreased energy transfer from NBD to Rhod. Successful reconstitution of SNARE proteins was validated by SDS-PAGE of the purified proteoliposomes (Supplementary Fig. 6).

Inspired by the stealth effect of poly(ethylene glycol) (PEG) on drug delivery vehicles[43], we first asked if coating liposomes with DNA tiles would serve a similar purpose, i.e. to impede fusion. 6HB-tile-coated v-liposomes (vT) were mixed with uncoated t-liposomes (t) in a 1:10 molar ratio and the fluorescence signal was monitored at room temperature for 4 h. Not surprisingly, substantial signal reduction was observed compared to uncoated v-liposomes (v + t), manifesting the steric shielding by tiles on membrane surface (Fig. 5a). More importantly, the suppressed fusion could be rescued when the coating was removed by TMSD, which affords for temporal regulation (Fig. 5a)[44]. Negative-stain TEM corroborated that many of the tile-studded liposomes were not associated with other liposomes, while more vesicle clusters appeared after coat removal (Supplementary Fig. 17). The shielding effect reached a maximum when tiles saturated liposome surfaces (Supplementary Fig. 18). No inhibition of fusion occurred when the tiles lacked handles,

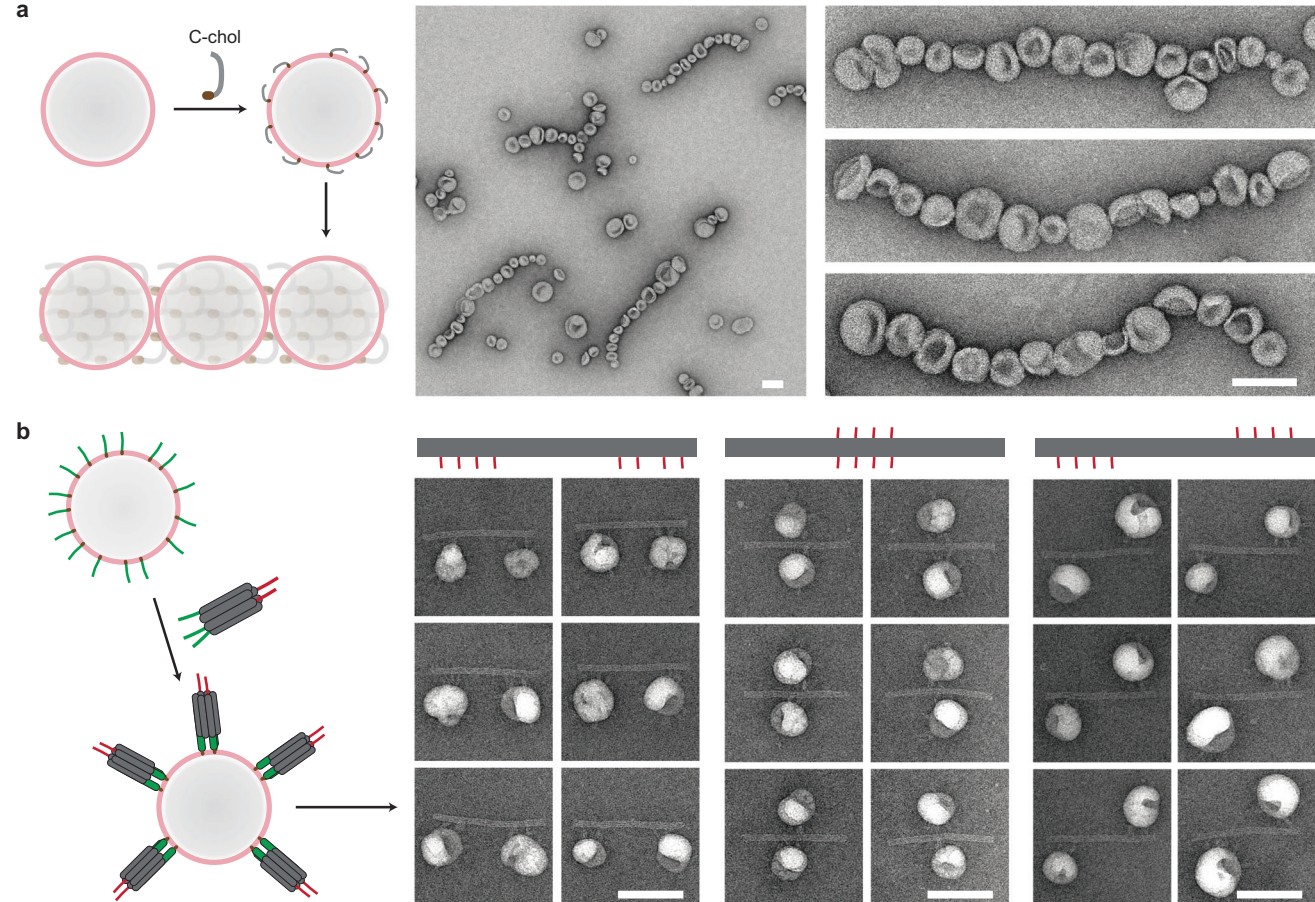

**Fig. 4 | Programmable higher-order organization of preformed liposomes by DNA self-assembly. a** Cartoon models (left) and negative-stain TEM images (right) of 1D liposome arrays templated by ribbon-forming C-tiles. **b** 6HB-tile-coated liposomes were placed at designated positions on an origami rod by handle hybridization between tile and origami. Cartoon model and six cropped TEM images are shown for each of the three distinct patterns. Scale bars: 100 nm.

further confirming fusion suppression arises from membrane surface modification (Supplementary Fig. 18).

After showcasing coating-hindered fusion, we next established that clustering of v- and t-liposomes together could promote fusion. DNA1-chol and DNA2-chol were incorporated into v-liposomes (v1) and t-liposomes (t2) respectively, then mixed before monitoring the fluorescence (Fig. 5b, see negative-stain TEM images in Supplementary Fig. 19). Notably, the initial fusion of v1 and t2 was slower than their uncoated counterparts (v + t), probably owing to some "protection" provided by the ssDNA coating, especially on the excess t-liposomes. At 100 min, a ssDNA complementary to both DNA1 and DNA2 (linker_12) was added for bridging. Immediately, the fusion rate increased dramatically, which eventually led to more complete fusion after 3 h than the uncoated control. Non-complementary linkers (linker_0) failed to affect fusion. A reaction with the same setup except for using protein-free liposomes (pf1+pf2) showed no fusion either before or after adding linker_12, suggesting that it was SNARE zippering, rather than DNA hybridization, that drove fusion. The role of DNA here was to mimic the tethering factors in vesicle trafficking[45], to pull two opposed membranes into proximity, thus aiding subsequent docking and fusion by SNAREs. We point out that our study differs from a previous report on DNA-accelerated SNARE-mediated membrane fusion[46], as here we focused on temporal control by introducing the linker strategy. Finally, we combined the two approaches above, i.e., tile coating and DNA linking, and achieved one-pot blocking and triggering of v- and t-liposome fusion (Supplementary Fig. 20).

## Discussion

Although DNA nanotechnology has proven its utility in directing the assembly of inorganic nanoparticles[47], convincing use of its potential for patterning similarly sized soft matter, especially biologically important lipid nanoparticles, have not been reported[48]. In this work we developed a versatile DNA-based method to coat, cluster, place, and control the fusion of sub-100-nm liposomes. Preformed liposomes were first decorated by cholesterol-modified oligonucleotides. Then, handle-equipped DNA nanostructures enabled vesicle functionalization with proteins and higher-order organization with precision. Many of the products reported here (e.g., liposome-ND core-satellite complexes, reversible liposome networks, 1D liposome arrays, liposome dimers, etc.) were only made possible thanks to the exceptional programmability of DNA. Moreover, the opposite effects of shielding and linking on SNARE-mediated membrane fusion reactions were exploited for temporal regulation.

The structural and functional manipulation of SUVs not only granted a capable and inspiring toolbox for drug carrier design, but also contributed a long-awaited asset to facilitate bottom-up synthetic biology[49,50]. For example, proteoliposomes barcoded by DNA tiles with distinct shapes can be used for biophysical studies of protein-mediated interactions between lipid bilayers. Reconfigurable DNA nanostructures may be incorporated into the system, to dynamically tune the inter-liposome spacing[38], ligand accessibility, and even membrane deformation[16]. Walkers or channels built from DNA might also be integrated to achieve vesicle transport and communication[51,52]. Last but not least, by scaling up the DNA-assembled structures[53], large unilamellar

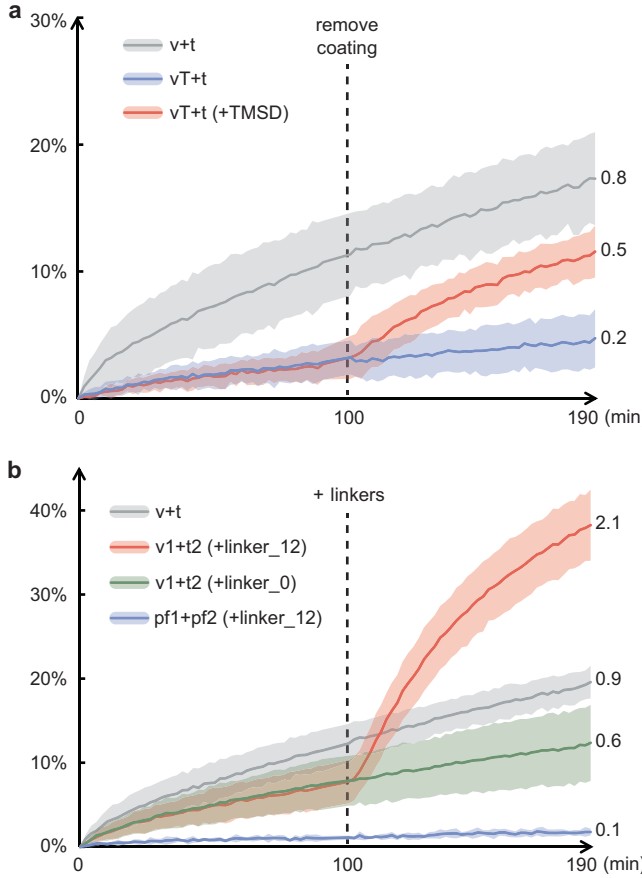

**Fig. 5 | Coating and/or linking liposomes for temporal control of SNARE-mediated membrane fusion.** Fusion kinetics are characterized by lipid mixing assays, where the v-liposome (v) contains 62% DOPC, 15% DOPE, 20% DOPS, 1.5% Rhod-PE, 1.5% NBD-PE and v-SNAREs, while the t-liposome (t) contains 65% DOPC, 15% DOPE, 20% DOPS and co-expressed t-SNAREs. **a** Tile-coated v-liposomes (vT) show suppressed fusion with uncoated t-liposomes (blue), which can be rescued after coat removal by TMSD (red) at 100 min. **b** Fusion between DNA1-tethered v-liposomes (v1) and DNA2-tethered t-liposomes (t2) can be greatly boosted by a complementary linker (linker_12) added at 100 min (red), but not by a non-complementary linker (green). No fusion occurs between ssDNA-tethered protein-free liposomes (pf1 and pf2) in the presence of the complementary linker (blue). Each curve in panel (**a**) shows the average of two batches of each sample; each curve in panel (**b**) shows the average of three batches of each sample; shading indicates the standard deviation. Estimated rounds of fusion, based on a previous calibration curve, are labeled at the end of each trace[17]. Source data are provided as a Source Data file.

vesicles (LUVs) (100–1000 nm) or even GUVs (>1 μm) could likewise be engineered by our method (see the coating and clustering of 50-nm or 200-nm extruded liposomes in Supplementary Fig. 21)[54], to mimic the configuration and dynamics of a wider variety of organelles.

## Methods

### Protein production
Membrane scaffolding protein (MSP), His-tagged v-SNARE (synaptobrevin-2) and t-SNARE (co-expressed syntaxin-1A and SNAP-25B) were produced and purified based on previous protocols[22,55].

### Liposome preparation
Liposomes were produced by dialysis-based detergent removal[42]. In brief, the lipid mixture with specified composition (1.5 μmol lipids in total) was nitrogen-dried and vacuum-dried in a glass tube, before resuspension with 500 μl *hydration buffer* (25 mM HEPES, 100 mM KCl,

1 mM TCEP, 5% glycerol, pH 7.4) containing 1.5% octyl β-D-glucopyranoside (OG) by vortex for 10 minutes. Another 1 ml *hydration buffer* was added dropwise to bring the OG concentration down to 0.5%. The solution was then transferred to dialysis tubing (MWCO 14 kDa, Sigma-Aldrich), and dialyzed against 1.5 liters *hydration buffer* at 4 °C overnight. Finally, the resultant sample was subjected to centrifugation (368,000 × *g* for 4 h at 4 °C, sw55Ti rotor, Beckman) in a three-layer iodixanol gradient (bottom layer: 1.5 ml sample + 1.5 ml 60% iodixanol; middle layer: 1 ml 20% iodixanol in *hydration buffer*; top layer: 300 μl *hydration buffer*), and 400 μl solution at the interface between the top and middle layer was collected, aliquoted, and stored at −80 °C.

For making proteoliposomes for the lipid mixing assay, 15 nmol of v-SNARE or 7.5 nmol of co-expressed t-SNARE was included in the original 500 μl *hydration buffer* before resuspension. The rest of the procedures were the same as the protocol above for protein-free liposomes.

### Nanodisc preparation
DNA-tethered v-SNARE-reconstituted nanodiscs (DNDv) were produced based on a previous study[22]. In brief, amine-modified ssDNA and cysteine-incorporated MSP were conjugated via a heterobifunctional linker, succinimidyl-4-(N-maleimidomethyl)cyclohexane-1-carboxylate (SMCC). Then, DNA-MSP conjugates, v-SNARE protein, and buffer-suspended lipids were mixed in a final concentration of 4 μM, 15 μM, and 300 μM respectively in 500 μl *hydration buffer* containing 0.06% n-dodecyl-β-D-maltoside (DDM). After 30 minutes of gentle shaking, 150 μl water-suspended Bio-Beads adsorbents (Bio-Rad) were added to remove DDM during an end-over-end sample mixing at 4 °C overnight. The solution was carefully collected and purified by a size-exclusion column (Superdex 200 increase 10/300) on a ÄKTA pure chromatography system (Cytiva), using *hydration buffer*.

### DNA tile design and preparation
6HB DNA tile was designed using caDNAno (cadnano.org) and revised from a previous version[56]. Desalted oligonucleotides were purchased from Integrated DNA Technologies (IDT) and used without further purification. To assemble each version of the tile, corresponding oligonucleotides were mixed equimolarly to a final concentration of 3 μM in *folding buffer* (5 mM Tris-HCl, 12.5 mM MgCl₂, 1 mM EDTA, pH 8.0), then annealed in a PCR thermocycler (Bio-Rad) by a linear cooling step from 95 °C to 20 °C for 2 h (−0.1 °C per 10 s). The products were purified by a size-exclusion column (Superdex 200 increase 10/300) on a ÄKTA pure chromatography system, using *hydration buffer* supplemented with 10 mM MgCl₂. Final tile concentration was determined by Nanodrop (Thermo Fisher Scientific) and calculated based on theoretical molecular weight.

### DNA origami design and preparation
DNA origami structures were designed using caDNAno. The hexagon origami was revised from a previous report[57]. To assemble each origami structure, the 8064-nt (for 180-nm rod) or 7308-nt (for 60-nm brick and 80-nm hexagon) scaffold strands (50 nM, M13-derived bacteriophages) were mixed with a selected pool of staple strands (300 nM each, IDT) in *folding buffer*, then annealed in a PCR thermocycler by a cooling step from 80 °C to 24 °C for 36 h[58]. The products were purified by rate-zonal centrifugation in glycerol gradients as described previously[59].

### Liposome coating and clustering by DNA nanostructures
Preformed liposomes were mixed with DNA-chol (HPLC-purified by IDT) in a DNA-chol-to-lipid molar ratio of 1:100, and incubated at 30 °C for 0.5 h. Pre-assembled 6HB tiles were then added in a tile-to-DNA-chol molar ratio of 1:10, and incubated at 30 °C for 1 h. No further purification was performed.

For origami-mediated liposome clustering, pre-assembled DNA origami was added to ssDNA-tethered liposomes in an origami-to-DNA-chol molar ratio of 1:60 and incubated at 30 °C for 1 h.

## Liposome functionalization and higher-order organization by DNA nanostructures

Tile-coated liposomes generated above were used subsequently in the following steps. For functionalization with streptavidin (SA), DNA-biotin (IDT) was added in a DNA-biotin-to-tile molar ratio of 2:1, and incubated at 30 °C for 0.5 h. Then SA (Roche) was added in a SA-to-biotin molar ratio of 2:1, and incubated at 30 °C for 1 h. The sample was subjected to isopycnic centrifugation in an iodixanol gradient for purification. For functionalization with v-SNAREs, premade DNDv was added in a DNDv-to-tile molar ratio of ~2:1, and incubated at 30 °C for 1 h. No further purification was performed for this sample. For origami-templated higher-order organization, pre-assembled origami rods were added in a rod-to-tile molar ratio of ~1:30, and incubated at 30 °C for 1 h. No further purification was performed.

## Liposome 1D array by C-tile assembly

Preformed liposomes were mixed with C-chol (HPLC-purified by IDT) in a C-chol-to-lipid molar ratio of 1:60, then annealed in a PCR thermocycler by a linear cooling step from 48 °C to 20 °C for 18 h (−0.1 °C per 4 minutes).

## Electron microscopy

For negative-stain EM, 5 µl sample (0.05 mM lipid concentration) was pipetted onto a glow-discharged formvar/carbon-coated copper grid (Ted Pella) and stained with 1% pH-adjusted uranyl formate for 1 min. Imaging was performed on a Talos F200C 200 kV scanning transmission electron microscope (Thermo Fisher Scientific). Images were acquired by TEM Imaging & Analysis (TIA). EM image measurements were performed in ImageJ 1.53.

For cryo-EM, 3 µl sample (0.8 mM lipid concentration) was pipetted onto a glow-discharged 200 mesh copper grid with lacey carbon film, blotted for 2 s, and plunge-frozen into liquid ethane. Frozen samples were imaged on a 300 kV Titan Krios cryo electron microscope, equipped with a high-brightness Field Emission Gun (x-FEG), a spherical aberration corrector, a Bioquantum energy filter and a K3 direct electron detector. Tomographic tilt series were recorded in super-resolution and dose-fractionation mode (movie frames). The calibrated pixel size was 2.06 Å per physical pixel (1.03 Å per super-resolution pixel) at a nominal magnification of 33kx. After pre-processing, the tilt series were aligned and reconstructed with AreTomo. Tomogram display and 3D modeling were performed using IMOD[60].

## Lipid mixing assay

Lipid mixing assays were based on fluorescence resonance energy transfer (FRET) between Rhod-PE and NBD-PE[42]. In brief, 30 µM Rhod/NBD-labeled v-liposomes (coated or uncoated) and 300 µM t-liposomes (coated or uncoated) were mixed (1:10 molar ratio) in 100 µl *hydration buffer* supplemented with 10 mM MgCl$_2$ (no MgCl$_2$ for Fig. 5b). Immediately after mixing, fluorescence (excitation: 475 nm, emission: 530 nm) was recorded every two minutes on a plate reader (BioTek Cytation 1, Agilent) at room temperature. Data were acquired by Biotek Gen5 (v3) and analyzed in Excel 2016 or Origin 2020. After 1.5 h, 2 µl ssDNA was added, and fluorescence was monitored for another 1.5 hours. Finally, 3 µl 20% OG was added, and fluorescence was measured for another 15 min. The final plot was normalized by the smallest (0%) and largest (100%) value of each sample recording.

## Statistics and reproducibility

Each experiment was repeated independently at least twice. Similar results were acquired and representative data were shown.

## Reporting summary

Further information on research design is available in the Nature Portfolio Reporting Summary linked to this article.

## Data availability

CaDNAno files and sequences of DNA nanostructures are available online. A cryo-ET video is available online. Source data are provided with this paper.

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

## Acknowledgements

We thank members of the Chapman Lab for helpful discussions and manuscript editing. This work was funded by the National Institutes of Health (grants MH061876 and NS097362 to E.R.C., grant R21NS123257 to Z.Z.). E.R.C. is an Investigator of the Howard Hughes Medical Institute. This article is subject to HHMI's Open Access to Publications policy. HHMI lab heads have previously granted a nonexclusive CC BY 4.0 license to the public and a sublicensable license to HHMI in their research articles. Pursuant to those licenses, the author-accepted manuscript of this article can be made freely available under a CC BY 4.0 license immediately upon publication.

## Author contributions

Z.Z. initiated the project, designed and carried out most of the experiments, analyzed the data, and prepared the manuscript. Z.F. carried out some of the experiments. X.Z., D.J., and Z.Y. performed cryo-EM studies. E.R.C. initiated the project, supervised the study, and prepared the manuscript. All authors reviewed and approved the manuscript.

## Competing interests

The authors declare no competing interests.
