## [Peer Review File · Nature Communications]

REVIEWER COMMENTS

Reviewer #1 (Remarks to the Author):

In the manuscript entitled "Functionalization and higher-order organization of liposomes with DNA nanostructures", the authors use DNA nanostructures to coat, cluster, and pattern sub-100-nm liposomes. The result showed successful construction of distance-defined liposome clusters, higher-order organization of liposomes, as well as the SNARE-mediated liposome fusion. The paper introduced a convenient and versatile method to build engineering premade vesicles both structurally and functionally. Still, a few questions need to be further addressed:

1. The liposome was generated using dialysis based detergent removal method. Was there a specific reason for the procedure choice, and would extruded liposomes work as well for this system? What is the size distribution of the raw liposomes in this work? Whether the liposome size changed after coating with the DNA materials? Quantitative analysis was highly recommended and expected.
2. In figure 2b, the cryo-ET images showed many liposomes with a darker inner circle, which seemed unusual. Are those products the multilamellar liposomes or overlaid particles? I have considered it to be the overlaid case until figure S9 said otherwise. While, if they are multilamellar vesicles, what is the reason of the different electron densities in the inner and outer layer spaces?
3. The distance between the distance-controlled liposome clusters mediated by different DNA nanostructure should be quantified and displayed.
4. The liposome fusion dynamics was monitored using plate reader. Is there any direct visual evidences with statistical analysis? What is the estimated rounds of fusion?

Reviewer #2 (Remarks to the Author):

This interesting work by Zhang et al reports a practical and programmable way to control the spatial arrangement of sub-100 nm liposomes and their fusion kinetics. Specifically, the authors used a combination of DNA origami structures, DNA tiles, and DNA oligonucleotides to control the connectivity, spacing, and merger of the liposomes and nanodiscs. Because liposomes are indispensable tools for studying membrane biophysics, this work is important and significant. Because most previous methods that use DNA to control liposome organization and dynamics either lack the level of precision reported here or require DNA-templated liposome formation, this work has sufficient elements of originality. The quality of data (EM images and fusion assays) are high and adequately support the claims.

I have a few comments for the authors to consider:

- (1) Although the liposome network may be difficult to purify because of the less-controlled network size, the dimeric liposomes should be amenable to enrichment by isopycnic centrifugation in a density gradient to remove the monomeric liposome and perhaps to enhance the liposome size homogeneity (Yang et al. Nat Chem 2021). While I understand that the purification step may not be important for proof-of-concept demonstration, authors may want to discuss this possibility.
- (2) It is interesting that the oligonucleotide linker could overcome the steric hindrance provided by the DNA tiles (Figure S20). Authors should specify the lengths of the oligonucleotides (DNA1-chol/DNA2-chol/linker_12) and analyze the effect of the oligonucleotide/linker lengths and the lengths/densities of the DNA tiles on the fusion kinetics.
- (3) The potential of the method presented here is enormous. For example, each type of the proteoliposomes can be labeled by a shape-barcoded DNA tile, thereby enabling detailed structural/biophysical study of their protein-mediated interactions. Reconfigurable DNA

nanostructures may be incorporated to dynamically tune the inter-liposome spacing, ligand accessibility, and even membrane deformation. In addition to serving as model membranes, size-controlled liposome clusters may also be useful for drug delivery. Authors may want to discuss some of these possibilities in the "Discussion and outlook".

Reviewer #3 (Remarks to the Author):

The manuscript by Zhang et al. demonstrated the programmable formation of vesicle (SUV) networks with DNA nanotechnology. Vesicles, including SUVs have already been functionalized with DNA in earlier studies by various groups (see comments). The linear arrangement of vesicles is nice (although not entirely new, see e.g. <https://www.nature.com/articles/nchem.2802>, Fig. 4), but like the entire manuscript it remains of a superficial descriptive manner. Statistical quantification of the observations are missing in the entire paper, the authors report merely observations, mainly based on TEM imaging. Overall, the TEM images are nice and helpful. With additional analysis (see comments below), the results are publishable, but in my opinion not on the level of Nature Communications.

Major comments:

1) The authors state that "Previously, simple DNA constructs like single-stranded (ss) or double-stranded (ds) DNA was interfaced with membranes to direct the assembly and/or drive fusion of liposomes, enabling programmable labeling of vesicles and control of energy input^{11,12}. Here we aim to expand this toolbox by introducing more sophisticated DNA architectures^{13,14} (...)" This is not a fair account of the literature given that also very complex and even functional origami structures were interfaced with SUVs, including DNA origami pores (e.g. <https://www.nature.com/articles/s41467-019-13284-1>), or reversible assembly of liposomes (<https://pubs.acs.org/doi/full/10.1021/acs.nanolett.6b01618>). It is simply not true that this was only done for GUVs.

2) The authors show TEM images which suggest that vesicles can be positioned at defined distances. These TEM images have to be analysed quantitatively by means of image analysis. The authors should provide mean and standard deviation of the placement and compare it to the theoretically expected value.

3) The 1 D liposome arrays are interesting, but also that has been done in slight variations in a paper cited by the authors (<https://www.nature.com/articles/nchem.2802>, Fig. 4) but also for pre-existing liposomes (e.g. <https://www.nature.com/articles/s41557-022-00945-w>). Therefore the statement has to be adapted: "It should be stressed that our method is applicable for pre-existing liposomes, while generation of liposome strings or dimers in previous studies required vesicle formed within DNA frames in situ^{35,36}. These two strategies could be viewed as complementary techniques for organizing vesicles at different stages of formation."

4) Again, for the 1D arrangement, analysis is missing entirely. On average, how many liposomes are arranged? How can this number be controlled, e.g. by the annealing time?

5) Figure 5 is interesting, but is this a one time measurement? Where are the error bars? How reproducible is it?

Minor comments:

1) The authors motivate their study with vaccine and drug-delivery applications. Why would vesicle networks in particular be beneficial here?

2) The authors report no unspecific binding of the DNA to the SUVs. This is expected to be highly buffer depended.

Reviewer #1:

In the manuscript entitled “Functionalization and higher-order organization of liposomes with DNA nanostructures”, the authors use DNA nanostructures to coat, cluster, and pattern sub-100-nm liposomes. The result showed successfully construction of distance-defined liposome clusters, higher-order organization of liposomes, as well as the SNARE-mediated liposome fusion. The paper introduced a convenient and versatile method to build engineering premade vesicles both structurally and functionally. Still, a few questions need to be further addressed:

We thank the reviewer for their summary and insightful comments. We address each of the questions that were raised, below.

1. The liposome was generated using dialysis based detergent removal method. Was there a specific reason for the procedure choice, and would extruded liposomes work as well for this system? What is the size distribution of the raw liposomes in this work? Whether the liposome size changed after coating with the DNA materials? Quantitative analysis was highly recommended and expected.

The reviewer asks about the liposome preparation method used in our study. Indeed, there are several ways to make SUVs. The most common methods are dialysis and extrusion, and the detailed protocols for each method vary somewhat between labs. We exclusively used dialysis in the current study for two reasons: (1) dialysis is the standard method for protein reconstitution in liposomes, and is therefore compatible with our application of SNARE-mediated membrane fusion reactions (Figure 5). (2) In our hands, liposomes formed by dialysis are more homogeneous than extruded liposomes and are therefore more amenable for patterning (Figure 4). However, this does not mean our system is only applicable to dialyzed liposomes. As a showcase, we extruded liposomes through filters with pore sizes of 50 and 200 nm, and successfully used 6HB DNA tiles to coat and cluster them. The results are shown in a new supplementary Figure S21, and are referenced in the ‘Discussion and outlook’ session in the manuscript where LUVs are mentioned.

“Figure S21. Coating and clustering extruded liposomes. a. Diameter of liposomes prepared by either dialysis (for detergent removal) or extrusion (through filters with pores of 50 or 200 nm) measured in negative-stain TEM images. More than 30 vesicles were measured for each sample. Mean diameter values are listed in the table (left) and are plotted in the bar graph (right); the error is the standard deviation. In general, extruded liposomes are larger and less homogeneous in size than dialyzed liposomes. **b.** Representative negative-stain TEM images show successful coating (left) or clustering (right) of two different-sized extruded liposomes (central) by 6HB DNA tiles. Scale bars: 200 nm.

Experimental protocol for liposome extrusion: Lipid mixture of 65% DOPC, 15% DOPE, and 20% DOPS (1.5 μmol total lipid) was nitrogen-dried and vacuum-dried in a glass tube, before rehydration with 500 μl buffer (25 mM HEPES, 100 mM KCl, pH 7.4) by

vortexing for 10 minutes. The solution was transferred into a 1.5 ml centrifuge tube and subjected to five freeze-thaw cycles between a liquid nitrogen bath (-196 °C) and a 37 °C water bath. Finally, the solution was sequentially extruded through polycarbonate filters with pore sizes of 50 or 200 nm, using a Mini Extruder (>40 passes each) at room temperature.”

The reviewer also asks whether liposome size changes after DNA coating. To address this, we measured the liposome size before and after 6HB tile coating in negative-stain (n.s.) TEM images. Interestingly, coated liposomes have slightly smaller diameters than uncoated ones (66 ± 14 vs. 72 ± 14 nm). Furthermore, the diameter of liposomes within tile-linked clusters is even smaller (58 ± 13 nm). Importantly, the vesicle size ‘returned’ to the original value after the clusters were disassembled by DNase (69 ± 14 nm). These findings suggest that the DNA tiles help liposomes to maintain their spherical shape (i.e., less flattened) on TEM grids. In this model, the ‘shrinking’ effect occurs only on imaging substrate, not in solution. Comprehensive cryo-EM studies will be useful to confirm this hypothesis in the future. We include and discuss these results in the new Supplemental Figure S10b.

“...**b**. Liposome diameters measured in negative-stain TEM images, showing that after clustering is reversed by DNase treatment, liposome sizes remain unchanged compared to the sizes before clustering (69 ± 14 vs. 72 ± 14 nm; $p = 0.29$ by two-tailed t-test). We note that liposomes are partially flattened on negative-stain TEM grids. For example, the measured diameter of liposomes within clusters is bigger in negative-stain TEM images than cryo-EM images (58 ± 13 vs. 53 ± 8 nm). Interestingly, DNA tiles, coated on the liposome surface, can help maintain the spherical shape of vesicles (i.e., they become less flattened) on TEM grids (66 ± 14 vs. 72 ± 14 nm; $p = 0.01$ by two-tailed t-test). More than 50 vesicles were measured for each sample. Mean diameter values are listed in the table (left) and are plotted in the bar graph (right); the error is the standard deviation.”

2. In figure 2b, the cryo-ET images showed many liposomes with a darker inner circle, which seemed unusual. Are those products the multilamellar liposomes or overlaid particles? I have considered it to be the overlaid case until figure S9 said otherwise. While, if they are multilamellar vesicles, what is the reason of the different electron densities in the inner and outer layer spaces?

The reviewer asks about the circles inside liposomes in cryo-EM images. We believe that they are nested rather than overlaid vesicles, because the inner and outer circle usually share the same focal plane. We are not sure how these liposome-in-liposome structures are formed, but it likely has to do with the dialysis process (during which OG is removed) and the purification step (i.e., isopycnic centrifugation). Regarding the greater electron density of these regions, we think it might be caused by the presence of iodine atom ($Z=53$) in the iodixanol molecule, which we used as medium for isopycnic

centrifugation. Potentially, some iodixanols diffuse into some of the liposomes, making the lumen darker by absorbing more electrons.

We included a discussion in supplementary Figure S9 as follows: *“We also note that higher electron density can be seen in some of the vesicles, which might be caused by the iodixanol used to purify liposomes via isopycnic centrifugation.”*

3. The distance between the distance-controlled liposome clusters mediated by different DNA nanostructure should be quantified and displayed.

We agree with the reviewer that quantitative analysis is desirable to validate the intended distance control. However, as the main characterization tool of this study, n.s. TEM suffers complications and difficulties in terms of measuring distances within liposome clusters, as follows: (i) it is difficult to distinguish individual vesicles and linkers within a cluster; (ii) a cluster in the 2D image is actually the projection of a partially collapsed 3D solution structure, so the measured distances will be smaller than the actual lengths; (iii) individual liposomes are flattened to some extent (see the new Figure S10), so the membrane-to-membrane distance will be underestimated; (iv) The resolution and contrast of n.s. TEM is limited, especially for lipid bilayers. In theory, cryo-EM doesn't suffer many of the problems above, but imaging all the structures by cryo is time-consuming and costly. As a compromise, we made the following quantification efforts, to strengthen our conclusions:

(1) Using cryo-EM images, we measured the inter-membrane distance within liposome clusters mediated by the exemplary 30-nm 6HB tiles (Figure 2). The value was 29 ± 2 nm, which matches the length of the linker tile (30 nm) well.

(2) Using n.s. TEM images, we measured the inter-membrane distance within liposome clusters mediated by the 30-nm tiles, 20-nm tiles, 60-nm rod origami, 80-nm hexagon origami, or 180-nm rod origami (Figure 3). We only measured the liposomes on the edge of the clusters, which are more distinguishable. In many cases, we also lowered the linker-to-lipid ratio to reduce the cluster size, which alleviates the above-mentioned problems (i) and (ii).

As expected, all the measured distances in n.s. TEM images are smaller than the theoretical value, including the same sample that has already been confirmed by cryo-EM in (1). Nevertheless, the results still revealed useful information about the design. First, longer DNA structures indeed lead to longer inter-membrane distances, fulfilling our goal of distance control. Second, interestingly, in all the cases except for the 80-nm hexagon origami, the measured inter-membrane distance is 12-15 nm shorter than the theoretical value. At least part of this difference ($\sim 58 - 53 = \sim 5$ nm) arises from vesicle flattening effect on n.s. TEM grids (see the new Figure S10b). Third, the 80-nm hexagon origami only separates liposomes by 48 nm on average, indicating the rigidity of this framework structure is not as high as the other rod-like structures.

To better integrate the relevant results and discussion in the manuscript, we made the following changes/rearrangements:

1. We added a bar graph showing distance measurement in Figure 3d. We added the following description in the manuscript:

“Small clusters also allow us to better distinguish individual liposomes and linkers, such that inter-membrane distance within a cluster can be measured in negative-stain TEM images. The results confirmed the trend of longer DNA nanostructures corresponding to larger inter-membrane distances, but in all cases the average distance was 12-15 nm shorter than design (Figure 3d). We attributed this discrepancy partly to vesicle flattening

(Supplementary Fig. S14). Another hexagon-shaped DNA origami was also assembled and used for liposome clustering, but the inter-membrane distance was not very well maintained probably due to its lower rigidity (Supplementary Fig. S14).”

“...d. Bar graph of inter-membrane distances measured in negative-stain (orange) or cryo (blue) EM images. The cryo measurement matches the design (grey) well, while the negative-stain measurement is shorter than the design partly due to vesicle flattening on substrate. Error bars are standard deviations.”

2. We combined all the results about the 80-nm hexagon origami and put them in Supplementary Figure S14b.

We added a cartoon model explaining vesicle flattening on n.s. TEM grids in Supplementary Figure S14a.

“Figure S14. a. Inter-membrane distances within DNA-mediated liposome clusters measured in EM images. Only distinguishable liposomes near the edge of the cluster were analyzed. More than 20 distances were measured for each sample. Average values and standard deviations are listed in the table. With the 30-nm 6HB tile as linker, the distances measured via cryo EM (29 ± 2 nm) match the design well, while the distances measured via n.s. TEM (16 ± 2 nm) are significantly shorter. Two major reasons account for this discrepancy: (i) Individual liposomes are flattened in n.s. TEM images (also see Figure S10b), thus shortening the gap between two neighboring liposomes (see cartoon model on the right); (ii) a cluster in the n.s. TEM image is actually the 2D projection of a partially collapsed 3D solution structure. Similarly, the measured inter-membrane distances in other liposome clusters (mediated by 20-nm tile, 60-nm rod origami, or 180-nm rod origami) in n.s. TEM are also 12-15 nm shorter than the theoretical value. b. Cartoon model and negative-stain TEM images of a hexagon DNA origami used for liposome clustering. The measured inter-membrane distance is 48 ± 10 nm, which is far less than the design (80 nm), indicating low rigidity of this framework structure. Scale bars: 100 nm.”

3. We moved the results regarding triangular liposome patterns in the original supplementary Figure S14 into supplementary Figure S13c of the revised manuscript.

4. The liposome fusion dynamics was monitored using plate reader. Is there any direct visual evidences with statistical analysis? What is the estimated rounds of fusion?

The reviewer wonders if there are other methods, especially methods providing direct visual evidence, to characterize liposome fusion dynamics. In our study, we did use n.s. TEM to visualize the products in many of the liposome fusion experiments (Figures S17-20). We found that a typical intermediate or final product for a liposome fusion reaction was a liposome cluster, likely mediated by v-/t-SNARE docking and/or partial zippering

(included in the figure legend of Figure S17). For the reasons mentioned above, we are unable to measure the liposome size within clusters in n.s. TEM, therefore no analysis on size change was performed. However, we emphasize that the main method we adopted in this study, i.e., the bulk fluorescence assay (based on the dilution of FRET donor/acceptor pairs on the v-SNARE liposomes, resulting in de-quenching of the donor), is the most commonly-used and widely-accepted assay to quantify SNARE-mediated liposome fusion dynamics, which has been established for 25 years ([10.1016/S0092-8674\(00\)81404-X](https://doi.org/10.1016/S0092-8674(00)81404-X)).

The reviewer also suggested that we estimate the rounds of fusion. Taking advantage of the standard curve published in a previous study using the same fluorescent lipid composition as ours ([10.1038/s41557-021-00667-5](https://doi.org/10.1038/s41557-021-00667-5)), we estimated the rounds of fusion at the end of each reaction (at 190 min), and labeled the value at the end of each curve in Figure 5.

“...Each curve shows the average of two duplicates of each sample; shading indicates the standard deviation. Estimated rounds of fusion, based on a previous calibration curve, are labeled at the end of each trace...”

Reviewer #2:

This interesting work by Zhang et al reports a practical and programmable way to control the spatial arrangement of sub-100 nm liposomes and their fusion kinetics. Specifically, the authors used a combination of DNA origami structures, DNA tiles, and DNA oligonucleotides to control the connectivity, spacing, and merger of the liposomes and nanodiscs. Because liposomes are indispensable tools for studying membrane biophysics, this work is important and significant. Because most previous methods that

use DNA to control liposome organization and dynamics either lack the level of precision reported here or require DNA-templated liposome formation, this work has sufficient elements of originality. The quality of data (EM images and fusion assays) are high and adequately support the claims.

We thank the reviewer for this summary, and we appreciate the positive feedback.

I have a few comments for the authors to consider:

(1) Although the liposome network may be difficult to purify because of the less-controlled network size, the dimeric liposomes should be amenable to enrichment by isopycnic centrifugation in a density gradient to remove the monomeric liposomes and perhaps to enhance the liposome size homogeneity (Yang et al. Nat Chem 2021). While I understand that the purification step may not be important for proof-of-concept demonstration, authors may want to discuss this possibility.

The review suggests using density gradient centrifugation to purify origami-directed liposome dimers. We appreciate this suggestion and performed the experiments on one of the dimer configurations. Indeed, isopycnic centrifugation (6%-30% iodixanol, 48000 rpm, 5 h) enriched liposome dimers in certain fractions (left image in the figure below). However, there are still individual liposomes coexisting in this fraction, probably because the density difference between tile-coated liposomes and origami-templated liposomes is not sufficient to obtain a perfect separation.

Inspired by the reviewer's suggestion, we also tried rate-zonal centrifugation (15%-45% glycerol, 48,000 rpm, 1 h), which separates particles by size rather than density. This time, almost all the individual liposomes are absent from the liposome dimer fraction (right image in the figure above). Note that the total volume of the medium gradient in

rate-zonal centrifugation is larger than isopycnic centrifugation (5 vs. 0.8 ml), which may also contribute to better separation in the former case.

We put the results of rate-zonal centrifugation in supplementary Figure S16a, as a demonstration of an optional purification step.

"...Excessive liposomes are required to saturate the binding sites on origami; optionally, they can be removed by rate-zonal centrifugation (15%-45% glycerol, 48,000 rpm, 1 h, 4 °C), as demonstrated using one of the configurations (bottom left). Scale bars: 200 nm."

(2) It is interesting that the oligonucleotide linker could overcome the steric hindrance provided by the DNA tiles (Figure S20). Authors should specify the lengths of the oligonucleotides (DNA1-cho/DNA2-cho/linker₁₂) and analyze the effect of the oligonucleotide/linker lengths and the lengths/densities of the DNA tiles on the fusion kinetics.

This is a very good point. The linker that connects two tiles is 63 bp, which translates to 21 nm. Each tile with cholesterol anti-handles at one end is 21 nm. Together, the total length of the DNA constructs after linking (named DC) is $(21 \times 2) + 21 = 63$ nm, which is already longer or on par with the liposome diameter (53 nm measured by cryo-EM). It was initially difficult to imagine two liposomes can fuse when such a rigid connector lies

between them, but the lipid mixing assay indeed revealed fusion. We propose the following explanation: During the formation of a DC-bridged liposome cluster, sometimes SNARE zippering occurs between a v-/t-liposome pair before full establishment of DC between all proteoliposomes. For example, in the figure below, two tile-coated v-liposomes (vT) and two tile-coated t-liposomes (tT) are connected in a network. However, the vT and tT at the bottom do not have DC between them (yet), making them ready to fuse via v- and t-SNARE interactions. We think there are many such cases during the formation of a liposome cluster, which boosted the bulk fusion rate by shortening the overall inter-liposome distance. We feel this is a likely explanation for our results, as it has been shown that any agent that helps bring v-SNARE and t-SNARE liposomes into proximity facilitates SNARE catalyzed fusion ([10.1038/nsmb.2075](https://doi.org/10.1038/nsmb.2075)).

If the hypothesis above is correct, the length of DC should have relatively minor effects on fusion, where longer DC leads to less fusion by increasing local inter-liposome distances (but still much shorter than the absence of DC). The role of the density of DC is a bit hard to predict, since more DC on the surface of liposomes means better clustering (good for fusion), but also leads to less scenarios as the figure shown above (bad for fusion). We feel that experimental validation for this hypothesis is beyond the scope of the current study.

We address this matter in the figure legend of Supplementary Figure S20, as follows:

"...We note that the total lengths of two linked tiles (~63 nm) is already longer or on par with the liposome diameter (~53 nm measured by cryo-EM). How such a DNA construct facilitates, rather than obstructs, membrane fusion is unclear. One likely explanation is that during the formation of linker-mediated liposome clusters, sometimes SNAREs between a v-/t-SNARE liposome pair can interact before the DNA constructs settle into place (see schematic on the upper right), locally promoting SNARE-mediated docking and zippering to increase the fusion rate. Hence, while the tiles can inhibit fusion between some v- and t-SNARE-bearing liposomes, our observations indicate that the net effect in the entire network is to promote their fusion...."

(3) The potential of the method presented here is enormous. For example, each type of the proteoliposomes can be labeled by a shape-barcoded DNA tile, thereby enabling detailed structural/biophysical study of their protein-mediated interactions. Reconfigurable DNA nanostructures may be incorporated to dynamically tune the inter-liposome spacing, ligand accessibility, and even membrane deformation. In addition to serving as model membranes, size-controlled liposome clusters may also be useful for drug delivery. Authors may want to discuss some of these possibilities in the "Discussion and outlook".

We thank the reviewer's optimistic view our method. We add many of these potential applications in the "Discussion and outlook" section as suggested:

"For example, proteoliposomes barcoded by DNA tiles with distinct shapes can be used for biophysical studies of protein-mediated interactions between lipid bilayers. Reconfigurable DNA nanostructures may be incorporated into the system, to dynamically tune the inter-liposome spacing, ligand accessibility, and even membrane deformation."

Reviewer #3:

The manuscript by Zhang et al. demonstrated the programmable formation of vesicle (SUV) networks with DNA nanotechnology. Vesicles, including SUVs have already been functionalized with DNA in earlier studies by various groups (see comments). The linear arrangement of vesicles is nice (although not entirely new, see e.g. <https://www.nature.com/articles/nchem.2802>, Fig. 4), but like the entire manuscript it remains of a superficial descriptive manner. Statistical quantification of the observations are missing in the entire paper, the authors report merely observations, mainly based on TEM imaging. Overall, the TEM images are nice and helpful. With additional analysis (see comments below), the results are publishable, but in my opinion not on the level of Nature Communications.

We thank the reviewer for this summary, and we appreciate the feedback.

We agree with the reviewer that quantitative analysis is desirable to strengthen our study. As a result, in this revision we performed quantitative measurements of: (1) vesicle size before and after coating/clustering, (2) vesicle distance within different clusters, (3) vesicle distance within the 1D string and dimeric configuration. In general, the measured values agreed well with the corresponding designs. We include these data in relevant main and supplementary figures.

Here we want to stress the novelty of our study, which is also indicated by the other two reviewers. We developed a powerful and versatile method to control the coating and spatial arrangement of sub-100 nm liposomes using DNA nanostructures. Previous work making 1D liposome strings relied on *in situ* liposome formation within DNA scaffolds ([10.1038/NCHEM.2802](https://doi.org/10.1038/NCHEM.2802)), while we focus on organizing vesicles produced by standard scaffold-free techniques (dialysis or extrusion). Previous work making DNA-mediated liposome aggregates only involved simple DNA constructs (i.e., ssDNA or dsDNA), while we harnessed the power of DNA tiles and origami to upgrade structural control to a much higher complexity level. We also demonstrate the utility of our approach to study biological problems by using it to regulate SNARE-mediated membrane fusion. We believe our method greatly elevates the potential of liposomes as delivery vehicles, model membranes, synthetic biology building blocks, etc.

Major comments:

1) The authors state that "Previously, simple DNA constructs like single-stranded (ss) or double-stranded (ds) DNA was interfaced with membranes to direct the assembly and/or drive fusion of liposomes, enabling programmable labeling of vesicles and control of energy input^{11,12}. Here we aim to expand this toolbox by introducing more sophisticated DNA architectures^{13,14} (...)" This is not a fair account of the literature given that also very complex and even functional origami structures were interfaced with SUVs, including DNA origami pores (e.g. <https://www.nature.com/articles/s41467-019-13284-1>), or reversible assembly of liposomes

(<https://pubs.acs.org/doi/full/10.1021/acs.nanolett.6b01618>). It is simply not true that this was only done for GUVs.

The reviewer mentioned a previous paper on DNA nanopores ([10.1038/s41467-019-13284-1](https://pubs.acs.org/doi/10.1038/s41467-019-13284-1)), to point out that there are already studies interfacing complex DNA nanostructures with SUVs. We completely agree, and in fact, there are also other papers using DNA tiles to sort liposomes ([10.1038/s41557-021-00667-5](https://pubs.acs.org/doi/10.1038/s41557-021-00667-5)) or sculpt liposomes ([10.1038/s41467-018-03198-9](https://pubs.acs.org/doi/10.1038/s41467-018-03198-9)). However, in this sentence/statement, we focus on the assembly and fusion of liposomes (“*to direct the assembly and/or drive fusion of liposomes*”), which indeed has only been achieved by ssDNA or dsDNA thus far. Moreover, we have avoided using the term ‘first time’ or ‘never’ throughout the manuscript, keeping in mind that all scientific research essentially builds on, or at least is inspired by, the work of others. Here we used the word ‘*expand*’ to acknowledge previous work and highlight our new direction on structural control. But to address the reviewer’s point, we add the above-mentioned three papers as new references (new ref 15-17).

The reviewer also mentioned a paper achieving DNA-mediated reversible assembly of liposomes ([10.1021/acs.nanolett.6b01618](https://pubs.acs.org/doi/10.1021/acs.nanolett.6b01618)). We actually cited this in our manuscript (original ref 27), where we disassembled liposome clusters by either DNase or TMSD (Figure 2c). To make things clearer, we revised this sentence:

“Previously, simple DNA constructs, like single-stranded (ss) or double-stranded (ds) DNA, were interfaced with membranes to direct the assembly/disassembly of liposome aggregates, and to drive vesicle fusion, in a programmable manner.”

In the end, the reviewer says ‘*It is simply not true that this was only done for GUVs*’. The only place we mentioned GUVs in the introduction session is this: “*Although existing techniques are capable of coating and associating giant unilamellar vesicles (GUVs), generic tools for functionalizing and patterning liposomes with diameters smaller than 100 nm have not been established.*” The key words here are ‘*generic tools*’, which are indeed missing for SUVs, and are what we strived to create in our study. We never claim anything that ‘*was only done for GUV*’, although indeed GUVs have not been spatially organized in ways that are as complex as our SUV patterns.

2) The authors show TEM images which suggest that vesicles can be positioned at defined distances. These TEM images have to be analysed quantitatively by means of image analysis. The authors should provide mean and standard deviation of the placement and compare it to the theoretically expected value.

We agree with the reviewer that quantitative analysis is desirable here. We measured the inter-membrane distance within liposome dimers in n.s. TEM images and organized the results in the new Figure S16b. The measured values generally agree well with the models.

b
“...b. Theoretical and measured inter-membrane distance between origami-templated dimeric liposomes. More than 30 distances were measured for each configuration. Average values, and standard deviations, are shown in both the table (left) and bar graph (right). In general, the liposome distances are consistent with the designs.”

We also mentioned this result in the main text: *“the measured inter-membrane distances are consistent with the designs (Supplementary Fig. S16).”*

We also measured inter-membrane distance within liposome clusters mediated by different DNA nanostructures. Please see the new Figure 3d and supplementary Figure S14a in our response to reviewer #1.

3) The 1 D liposome arrays are interesting, but also that has been done in slight variations in a paper cited by the authors (<https://www.nature.com/articles/nchem.2802>, Fig. 4) but also for pre-existing liposomes (e.g. <https://www.nature.com/articles/s41557-022-00945-w>). Therefore the statement has to be adapted: “It should be stressed that our method is applicable for pre-existing liposomes, while generation of liposome strings or dimers in previous studies required vesicle formed within DNA frames in situ^{35,36}. These two strategies could be viewed as complementary techniques for organizing vesicles at different stages of formation.”

The reviewer mentioned a previous paper ([10.1038/s41557-022-00945-w](https://doi.org/10.1038/s41557-022-00945-w); Zhan et al. 2022), and suggests that arranging pre-existing liposomes into a 1D array has been achieved. However, the most relevant data in that paper, with regard to our study, is in Figure 4 where there is an EM image showing multiple vesicles randomly attached to entangled DNA tubes (see left figure below). In contrast, in our study, tightly packed liposome strings are templated by DNA ribbons (right figure below), which are much more ordered and controlled. The Zhan paper is about vesicle movement on a DNA track; we actually cited this study in our manuscript (original ref 48), where we envisioned integrating our system with “walkers” to achieve vesicle transport in the future. In brief, from our understanding, making ordered liposome strings is neither the intention nor the product of the Zhan paper, therefore it doesn’t weaken the novelty of our method. In the statement that the reviewer mentioned, we did not claim this has not been done before. We simply acknowledge previous work making liposome strings ([10.1038/NCHEM.2802](https://doi.org/10.1038/NCHEM.2802)) or dimers ([10.1038/s41589-019-0325-3](https://doi.org/10.1038/s41589-019-0325-3)) using liposomes formed *within* DNA scaffolds, and highlight the differences in comparison to our method (liposome formation *without* DNA scaffolds). We think it’s a fair and important statement. But to clear any confusion, we add the word ‘ordered’ in the following statement:

“It should be stressed that our method is applicable for pre-existing liposomes, while generation of ordered liposome strings or dimers in previous studies required vesicle formed within DNA frames in situ.”

Figure 4b from 10.1038/s41557-022-00945-w

Figure 4a from this paper

4) Again, for the 1D arrangement, analysis is missing entirely. On average, how many liposomes are arranged? How can this number be controlled, e.g. by the annealing time?

We agree with the reviewer that quantitative analysis is desirable here. We have counted the number of liposomes in individual strings from n.s. TEM images and plotted a histogram. The peak shows up at 5 liposomes per string under the conditions that we used.

The reviewer asks about tuning string length by changing the annealing time. Because the current annealing time is already as long as 18 hours, we suspect that even longer times would only affect the products marginally. Instead, we tested different starting temperatures (54/48/40 °C) for the 18-hour annealing (ending temperature kept at 20 °C). It seems that higher starting temperature leads to strings composed of more liposomes (9 vs. 5 liposomes per string), but includes more aggregates. 48°C-20°C-18h is an optimal annealing protocol for enriching liposome strings.

We show the relevant EM images and plots in the new supplementary Figure S15.

“d. Effect of starting temperature in an 18-h annealing (ending temperature kept at 20 °C) when only C-cho is used. A representative TEM image and a histogram of liposome counts in a string are shown for each starting temperature (54 or 48 or 40 °C). In all cases, 1D liposome strings along with isolated liposome individuals and clusters coexist in the products. Apparently, higher starting temperature (e.g., 54 °C) leads to strings composed of more liposomes (9 vs. 5 liposomes per string), along with more aggregates. 48°C-20°C-18h is an optimal annealing protocol for enriching liposome strings.”

5) Figure 5 is interesting, but is this a one time measurement? Where are the error bars? How reproducible is it?

The line charts in Figure 5 showed representative curves acquired in a single experiment. Showing representative curves is a common and acceptable way of presenting liposome fusion kinetics in the study of SNAREs (e.g. [10.1038/nsmb.3141](https://doi.org/10.1038/nsmb.3141)), assuming the data are reproducible. In our hands, the absolute fusion rate of each condition may vary up to 20% when different batches of materials (proteins, DNA nanostructures, liposome preparations, et al.) are used, but the trend between samples (relative difference) always remains the same. To include error bars as the reviewer suggests, we used the data from two repeats for each sample, and plotted the error bars as shaded area in the new Figure 5.

Minor comments:

1) The authors motivate their study with vaccine and drug-delivery applications. Why would vesicle networks in particular be beneficial here?

We envision drug delivery applications mainly for our liposome coating method, as surface modification of nanocarriers can improve their performance enormously (overcoming biological barriers, increasing target property, managing circulation time, etc.; [10.1021/acsnano.2c02347](https://doi.org/10.1021/acsnano.2c02347)). As for liposome networks, although it is unclear how clustering affects these properties, as reviewer #2 points out, reversible or spacing-switchable clusters could be used for tuning ligand accessibility, therefore providing a way of stimulus-responsive targeting/release.

2) The authors report no unspecific binding of the DNA to the SUVs. This is expected to be highly buffer depended.

We are aware of previous reports regarding unspecific binding of DNA to liposomes (e.g., 10.1021/jacs.1c00166, 10.1039/d2nr05368c), some of which we have cited in our manuscript. As the reviewer points out, such interactions are highly buffer dependent, as well as lipid composition and DNA nanostructure dependent. Our liposomes are composed of liquid-phase bilayers (65% DOPC, 15% DOPE, 20% DOPS) that do not favor unspecific binding of DNA. Similar lipid composition and buffer systems have been used before in the context of DNA nanostructures interacting with membrane, while no unspecific binding was reported (e.g. 10.1038/NCHEM.2802, 10.1038/s41557-021-00667-5).

To acknowledge the potential unspecific binding under other conditions, we add '*in the current system*' when unspecific binding is mentioned in the main text:

“No attachment was observed for tiles without handles or with three non-complementary handles, ruling out the presence of unspecific binding due to electrostatic interactions in the current system (Supplementary Fig. S3).”

REVIEWER COMMENTS

Reviewer #1 (Remarks to the Author):

The authors have properly addressed all the questions from me and other reviewers. The manuscript seems in a good shape for publication in Nature Communications now.

Reviewer #2 (Remarks to the Author):

The revised manuscript has addressed the points raised by me and other reviewers. In my opinion, this is an interesting study with very high quality and should be published without delay.

Reviewer #3 (Remarks to the Author):

The statistical representations did definitely improve the work. I still believe that it is not sufficient to report two traces in support of the fusion events. In their answer the authors state that the variation is 20 %. I would like to see the data that supports this statement. Other than that I am happy with the revision.

Reviewer #3 (Remarks to the Author):

The statistical representations did definitely improve the work. I still believe that it is not sufficient to report two traces in support of the fusion events. In their answer the authors state that the variation is 20 %. I would like to see the data that supports this statement. Other than that I am happy with the revision.

We appreciate the reviewer's rigor regarding data quality and reproducibility.

The reviewer indicates that two traces of each sample are not sufficiently representative. Therefore, in this revision, we use three or four traces (the exact number of traces are specified in the legend for Fig. 5) of each sample to better determine the median and standard deviation (SD). Because we now combine data that were obtained using different batches of materials (proteins, DNA nanostructures, liposome preparations, etc.; the number of batches is also specified in the legend for Fig. 5), the SD is larger than when a single batch of materials was used. Hence, we generated new line plots (median), with the shaded area indicating the error (SD), in a new version of Figure 5, as shown below. Note that we removed the trace for tile-coated liposome fusion treated by DNase (originally green trace in 5a), as we realized that DNase may not function properly due to lack of Mg^{2+} in the buffer in this particular experiment. DNase treatment, in all of the other experiments in our study, should function normally as there is Mg^{2+} in all the buffers for those experiments. TMSD doesn't need Mg^{2+} to function, so it's still an appropriate method to remove the coating and promote fusion (red trace in 5a).

“Figure 5... Each curve in panel (a) shows the average of three repeats from two batches of each sample; each curve in panel (b) shows the average of four repeats from three batches of each sample; shading indicates the standard deviation...”

The reviewer wants to see the evidence supporting our statement that data from each sample always varied less than 20%. We have attached an excel file (source data for Figure 5 for reviewer 3.xlsx) containing all the source data for the new Figure 5. While the variation is indeed ~20% or less within batches, in our broader analysis that expands the data set to include trials from other batches of materials, we found that the variation was sometimes greater than 20% between batches. Below we provide a table summarizing the final percentages of fusion completion of all the samples in Figure 5 in different trials. Note that here all the percentages mean normalized fusion level after 3 hours, not the percentage of variation between samples. The percentage of variation can be calculated using the values in the table. For example, trial 2 and trial 3 of vT+t (+TMSD) differs $(13.2\% - 10.1\%) / 13.2\% = 23\%$.

Sample	batch 1		batch 2	batch 3
Fig. 5a	trial 1	trial 2	trial 3	trial 4
v+t	19.9%	19.9%	14.7%	
vT+t	5.9%	6.8%	3.1%	
vT+t (+TMSD)	12.8%	13.2%	10.1%	
Fig. 5b				
v+t	19.9%	19.9%	17.5%	21.3%
v1+t2 (+linker_12)	40.9%	42.8%	33.6%	39.2%
v1+t2 (+linker_0)	14.9%	16.0%	7.2%	14.4%
pf1+pf2 (+linker_12)	1.1%	1.5%	2.1%	1.9%

Basically, fusion completion in batch 2 is systematically smaller than for batches 1 and 3, and the difference is larger than 50% in some cases (e.g., vT+t and v1+t2 (+linker_0)). This is probably caused by differences in reconstitution efficiency, pipetting variation, differences in the activity of some of the reagents, etc.

As we pointed out last time, the trend between samples (i.e., relative value) always remains the same in every batch, which is sufficient to demonstrate the effect of coating and linking on fusion. This is also the reason why we used data from the same batch to make plots in the previous versions of our study: the effect of DNA manipulation and intervention could be more clearly illustrated. We also note that showing representative curves is a common and acceptable way of presenting liposome fusion kinetics in the study of SNAREs (e.g., 10.1038/nsmb.3141, 10.1002/anie.201506844). Regardless, we hope we have fully addressed the concerns, raised by the referee, by expanding our data set to include trials using different batches of materials. And while the error is a bit larger than 20% (when using different batch materials to address a question), our conclusions still hold, and the rigor of our study is enhanced.

In compliance with Nature Communications' policy of data availability and transparency, we also submit an excel file (source data.xlsx) containing all the source data of lipid mixing assay traces and EM image measurement, for all the relevant main and supplementary figures, to Figshare.

REVIEWERS' COMMENTS

Reviewer #3 (Remarks to the Author):

The authors have fully adressed my concerns. Thank you.